# Safety and Protection Measures of Underground Non-Coal Mines with Mining Depth over 800 m: A Case Study in Shandong, China

Li Cheng [1], Qinzheng Wu [1], Haotian Li [2], Kexu Chen [1], Chunlong Wang [1], Xingquan Liu [1], Xuelong Li [2,*] and Jingjing Meng [3]

1   Deep Mining Laboratory of Shandong Gold Group Co., Yantai 264000, China
2   College of Energy and Mining Engineering, Shandong University of Science and Technology, Qingdao 266590, China
3   Department of Civil, Environmental and Natural Resources Engineering, Luleå University of Technology, 971 87 Luleå, Sweden
*   Correspondence: lixl@sdust.edu.cn

**Abstract:** With the increase in mining depth, the risk of ground pressure disasters in yellow gold mines is becoming more and more serious. This paper carries out a borehole test for the pressure behavior in a non-coal mining area with a mining depth of more than 800 m in the Jiaodong area. The test results show that under a depth of 1050 m, the increase in the vertical principal stress is the same as the increase in the minimum horizontal principal stress, which is about 3 MPa per 100 m. When the depth increases to 1350 m, the vertical principal stress increases by about 3% per 100 m, and the self-weight stress and the maximum horizontal principal stress maintain a steady growth rate of about 3 MPa per 100 m. In addition, based on the test results, the operation of the ground pressure monitoring system in each mine is investigated. The investigation results show that in some of the roadway and stope mines with depths of more than 800 m, varying degrees of rock mass instability have occurred, and a few mines have had sporadic slight rockbursts, accounting for about 5%. There was a stress concentration area in the lower part of the goaf formed in the early stage of mining, and slight rockburst phenomena such as rock mass ejection have occurred; meanwhile, the area stability for normal production and construction was good, and there was no obvious ground pressure. This paper compares the researched mines horizontally as well as to international high-level mines and puts forward some suggestions, including: carrying out ground pressure investigations and improving the level of intelligence, which would provide countermeasures to balance the safety risks of deep mining, reducing all kinds of safety production accidents and providing a solid basis for risk prevention and supervision.

**Keywords:** non-coal mine; great mining depth; ground pressure disaster; safety investigation

## 1. Introduction

China is one of the primary gold-producing areas in the world. As a critical non-coal mineral resource, gold accounts for a large proportion of the national economy [1,2]. The Jiaodong area of Shandong Province [3], which is the largest gold producing area in China, produces many world-class large gold deposits (more than 100 tons) and many small- and medium-sized gold deposits [4,5], including the Zhaoyuan-Laizhou, Penglai-Qixia, and Muping-Rushan gold deposits. Among them, the gold resources of the Zhaoyuan-Laizhou metallogenic area account for more than 80% of the total reserves of Shandong Province. Sai et al. inferred the southward extension of the Zhaoyuan-Pingdu gold belt under the Quaternary cover area via comprehensive multivariate information, such as geological exploration, geophysical, and geochemical data [6]. Cheng et al. established metallogenic

models of altered rock type gold deposits and quartz vein type gold deposits in the fracture zone, which verifies the great prospecting potential in the Jiaodong area [7].

With the increase in mining depth, the risk of ground pressure disasters is becoming more serious. Defining the distribution of the in situ stress field can effectively and accurately measure, prevent, and control ground pressure disasters. At present, in situ stress surveys have been carried out in more than thirty countries globally [8–10], but the application of technical research in China lags. The main task of detecting underground stress fields in deep mines is determining the activity intensity and the mode of ground stress, as well as the direction of the main stress and its fundamental mechanism of changing with depth [11]. Liu et al. obtained the rockburst tendency evaluation results by drilling cores in the Sanshandao gold mine. It was proposed that the corresponding intensity grade would increase with the mining depth [12]. Qin et al. proposed that in situ stress measurements should select measuring points according to rock mass structures and should avoid interference sources, such as stress fields and unstable regions, to ensure the authenticity of the stress data. Their theory played a guiding role in the selection of the Xiling mining area in Sanshandao as the drilling location [13]. Based on the measurements of the in situ stress field of a specific mine in Jiaodong, it was found that the stress in the deep mining area is mainly horizontal tectonic stress [14], and proper support was suggested to be provided in time. Li et al. drilled fifty-three measuring points in the Sanshandao, Xincheng, and Linglong gold mines [15], and they found that the maximum horizontal stress increases with depth. Du et al. improved the traditional coring technique and predicted the potential location of rockbursts and their intensity in future mining activities of the Sanshandao gold mine using a theoretical analysis and numerical simulation [16].

The appearance of ground pressure will have an impact on production safety, so some mines have adopted new support schemes [17–19]. Jinqingding mine of the Jinzhou Mining Group is gradually replacing its rigid support with flexible support [20,21], and advanced grouting is used in local broken areas. In the Sanshandao gold mine, resin bolts are widely used in deep support to replace pipe joint bolts, anchor-shotcreting nets, and piercing belts during excavation [22–24]. These measures effectively reduce the damage caused by ground pressure, but at the same time, their high cost makes it difficult to popularize them in all mines. We aim to find a more reasonable support scheme. In addition, numerical simulation software, such as COMSOL and FLAC3D, can establish mathematical models more intuitively and promote the research of mine safety production [25–28]. Some studies have used the constitutive model or time–frequency transformations to study rock mass and coal-rock assemblage, and an instability prediction method has been established [29–31]. However, the above research does not consider the problem of high temperature, ignores the complex physical and chemical interactions between rock masses, and is not suitable for deeper mining.

Compared with foreign advanced mines, there are some common problems in domestic gold mines, such as a lack of rock mechanics data [32], unsuitable support in some areas [33], and so on. Based on previous studies, this paper found some new problems, such as inadequate mine pressure monitoring, imperfect ventilation systems, and so on. This paper focuses on the present situation and the existing problems of ground pressure, and puts forward some suggestions for preventing ground pressure disasters and ensuring safe productions. In addition, a basic microseismic monitoring and analysis method is determined through in situ stress measurement and analysis, and a long-term construction strategy of the mine is proposed, which provides a feasible scheme for the construction of a green, intelligent mine.

## 2. Basic Situation of Mines

In regards to their geographical location, the nineteen underground non-coal mines investigated in this study are distributed in Yantai and Weihai. There is one in Muping, Yantai, three in Penglai, one in Longkou, twelve in Zhaoyuan, one in Laizhou, and one in Rushan, Weihai. Sixty percent of the mines are located in Zhaoyuan and fifteen percent

in Penglai. The geographical location of each mine is shown in Figure 1. In terms of ownership, six belong to the Shandong Gold Group, one belongs to a local state-owned enterprise, four belong to the Shandong Zhaojin Group, four belong to the Shandong China Mining Group, and the remaining four belong to collective enterprises or private enterprises. Collective city-owned mines accounted for 37%, and provincial state-owned mines accounted for 32%. When classifying the mines by their production and construction stages, one is newly built. Five have completed reconstruction and expansion as well as the completion and acceptance of their safety facilities. Thirteen are in stages of reconstruction and expansion during production, accounting for 68%. Among them, the safety facilities of eleven mines of the project have passed acceptance inspections, and two have not yet had complete acceptance of their safety facilities. In terms of their states of production and construction, twelve are in normal production and construction, and six are in policy shutdown or accident shutdown, accounting for 30%. The situation is shown in Table 1. Pvso means provincial state-owned, Cco means collective city-owned, Tse means township enterprises, Pve means private enterprise, Pdt means production, Cst means construction, Nm means normal construction, Plc std means policy shutdown, and Acd std means accident shutdown.

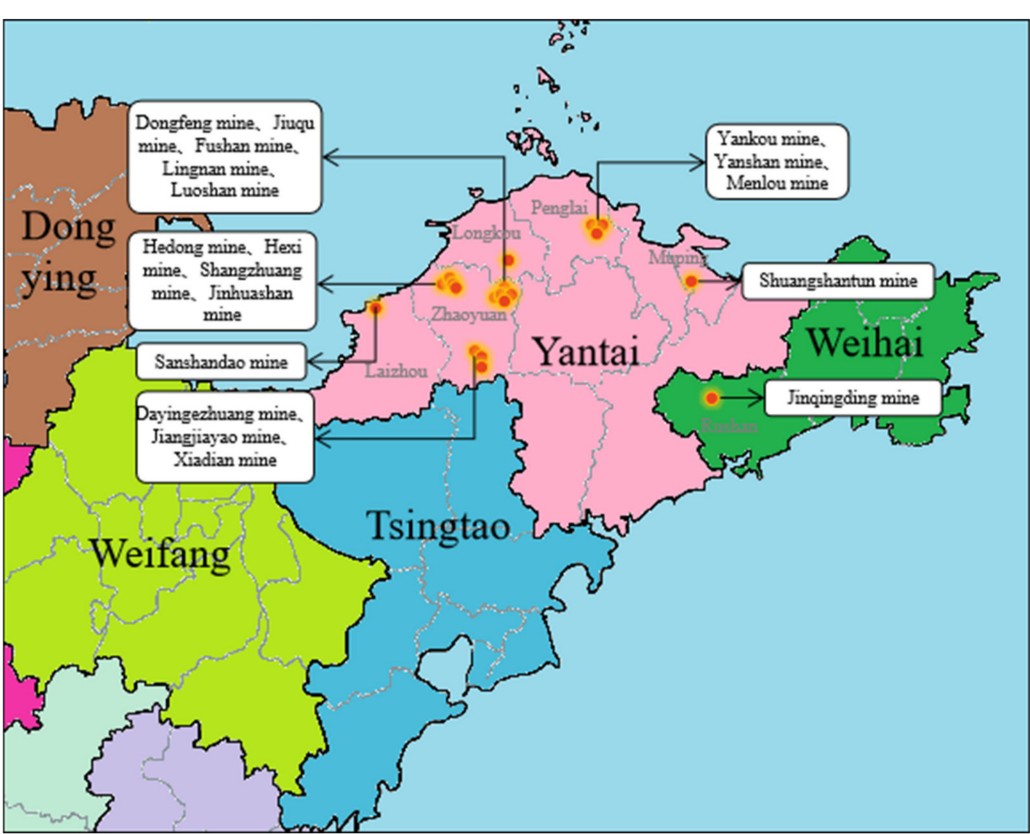

**Figure 1.** Geographical map of nineteen mines with mining depths of more than 800 m.

**Table 1.** List of location, ownership, stage, and status of nineteen mines with mining depths of more than 800 m.

| No | Abbreviation | Location | Owner-Ship | Stage | Status |
|----|----|----|----|----|----|
| 1 | Yantai Baihen Shuangshantun Mine | Muping | Pve | Cst | Nm |
| 2 | Shnajin Jinchuang Yanshan Mine | Penglai | Pvso | Pdt+Cst | Nm |
| 3 | Shanjin Jinchuang Yankou Mine | Penglai | Pvso | Pdt+Cst | Nm |
| 4 | Penglai Menlou Mine | Penglai | Pve | Pdt+Cst | Acd std |

| No | Abbreviation | Location | Owner-Ship | Stage | Status |
|----|-------------|----------|-----------|-------|--------|
| 5 | Longkou Jintai Damoqujia Mine | Longkou | Pve | Pdt+Cst | Plc std |
| 6 | Shanjin Linglong Jiuqu Mine | Zhaoyuan | Pvso | Pdt | Plc std |
| 7 | Shanjin Dongfeng Mine | Zhaoyuan | Pvso | Pdt | Nm |
| 8 | Zhaojin Xiadian Mine | Zhaoyuan | Cco | Pdt+Cst | Nm |
| 9 | Zhaojin Dayingezhuang Mine | Zhaoyuan | Cco | Pdt+Cst | Nm |
| 10 | Zhaojin Shangzhuang Mine | Zhaoyuan | Cco | Pdt+Cst | Nm |
| 11 | Zhaojin Hedong Mine | Zhaoyuan | Cco | Pdt | Nm |
| 12 | Zhongkuang Lingnan Fifth Mine | Zhaoyuan | Cco | Pdt+Cst | Plc std |
| 13 | Zhongkuang Luoshan Forth Mine | Zhaoyuan | Cco | Pdt+Cst | Plc std |
| 14 | Zhongkuang Fushan Dongfeng Mine | Zhaoyuan | Cco | Pdt | Plc std |
| 15 | Zhaoyuan Jiangjiayao Mine | Zhaoyuan | Tse | Pdt+Cst | Nm |
| 16 | Zhaoyuan Hexi Mine | Zhaoyuan | Tse | Pdt | Nm |
| 17 | ZhaoyuanLingshan Jinhuashan Mine | Zhaoyuan | Tse | Pdt+Cst | Nm |
| 18 | Shanjin Sanshandao Mine | Laizhou | Pvso | Pdt+Cst | Nm |
| 19 | Shanjin Jinzhou Jinqingding Mine | Rushan | Pvso | Pdt+Cst | Nm |

## 2.1. Geological Condition

The nineteen mines investigated are all gold mines, and all of them are distributed in the Jiaodong area. In terms of geotectonic location, the Jiaodong area is located on the southeastern margin of the North China Craton, bounded by the Tan-Lu fault zone in the west and the northern segment of the Sulu ultra-high-pressure metamorphic belt in the east (Figure 2). It is China's largest gold-producing area, producing several world-class large gold deposits (more than 100 tons) and many small- and medium-sized gold deposits. There are mainly three gold metallogenic areas in the Jiaodong area: Zhaoyuan-Laizhou, Penglai-Qixia, and Muping-Rushan. Most of the investigated mines are located in the Zhaoyuan-Laizhou metallogenic area. The gold reserve in this area accounts for more than 80% of the gold reserves in Shandong Province.

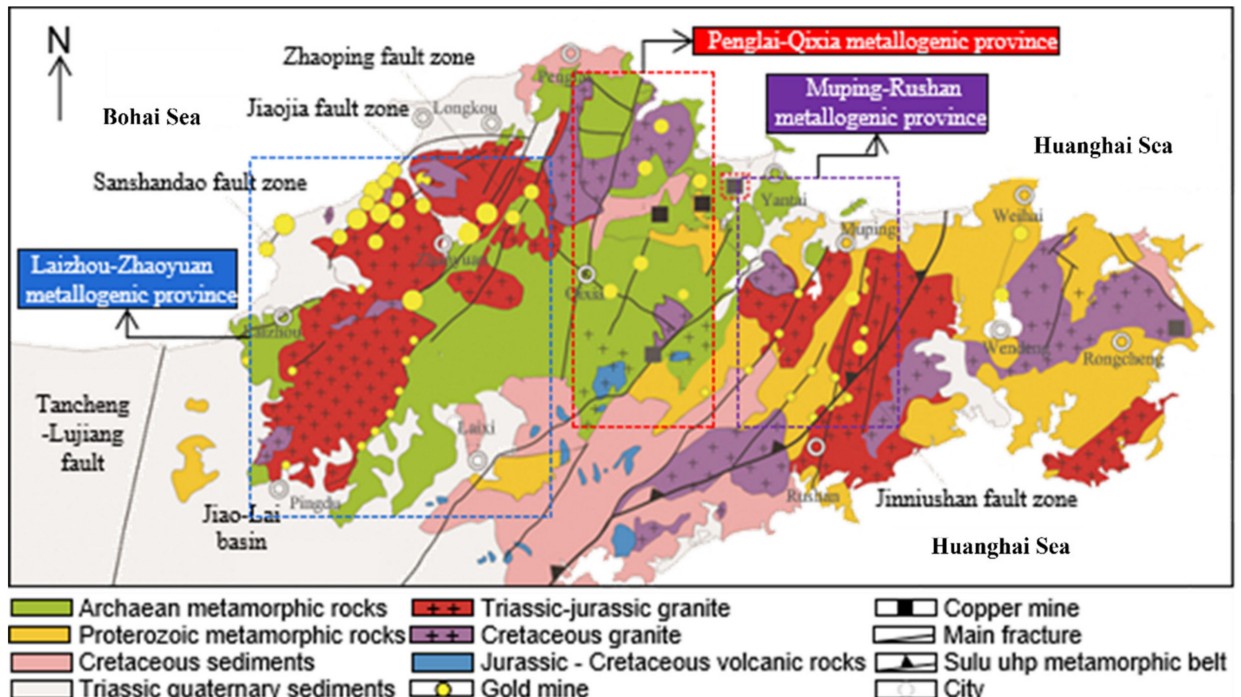

**Figure 2.** Regional geological map of Jiaodong area.

The strata exposed in the Jiaodong area are mainly Upper Archean Jiaodong Group, Lower Proterozoic Jingshan Group, Fanzishan Group, Upper Proterozoic Penglai Group, Mesozoic Jurassic and Cretaceous, and the Tertiary and Quaternary of Cenozoic. Magmatic rocks are widely distributed in Jiaodong, and intrusive rocks are distributed everywhere, accounting for more than half of the bedrock area (Figure 3). The magmatic rocks in the area can be divided into five types: exquisite granite, Guojialing granite, Aishan granite, Kunyushan granite, and Sanfoshan granite.

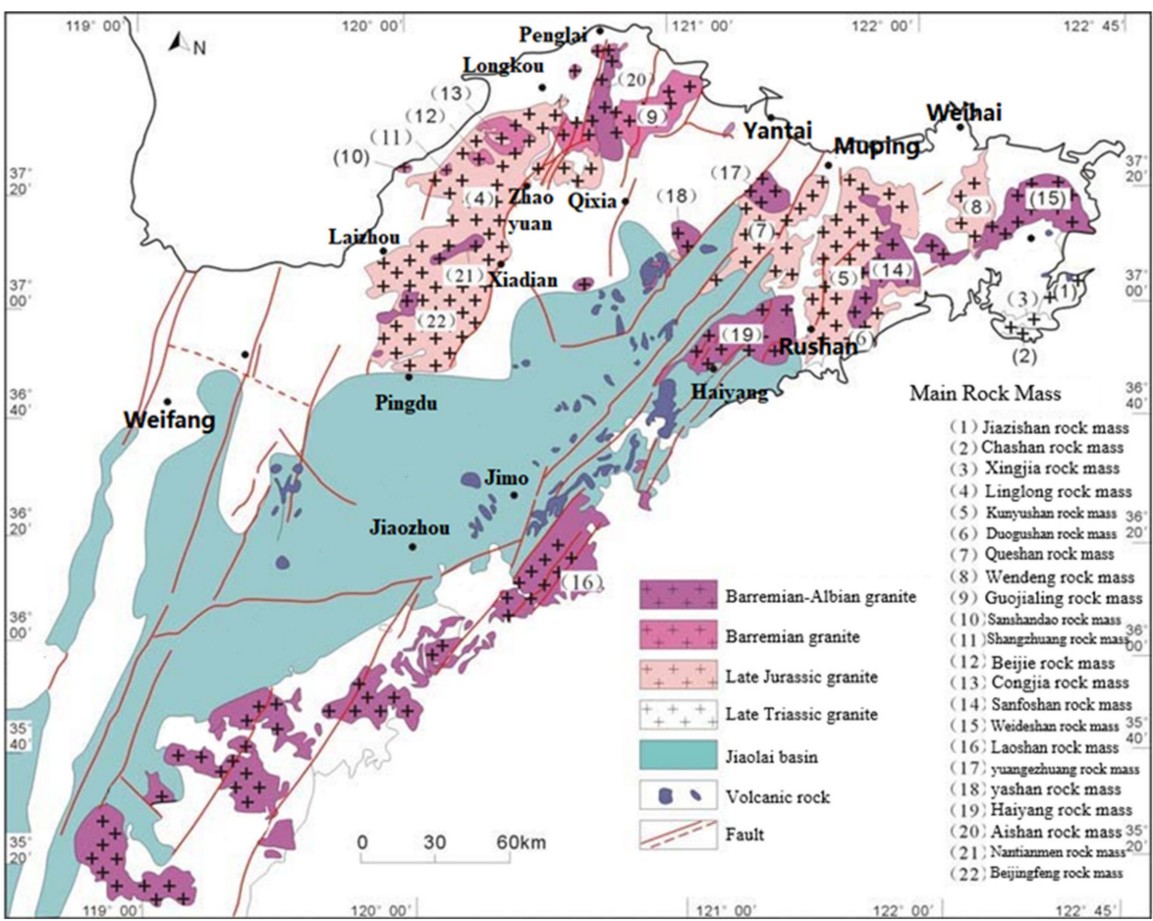

**Figure 3.** Magmatic rock distribution map in the Jiaodong area.

The development of fault structures characterizes the basic structural framework of Jiaodong. Its main fault zones are along the NE and NNE directions. The secondary fault zones are along the NW and SN directions (Figure 4). There are seven major faults in the region: the Yishu fault (Shandong section of Tanlu fault), Wulian-Jimo-Muping fault, Zhao-Ping fault, Long-Lai fault, San-Cang fault, Mu-Ru fault, and the Peng-Qi fault. Among them, the the Yishu fault and the Wulian-Jimo-Muping fault zone are the first-order structures in the region. In contrast, the Zhao-Ping fault, Long-Lai fault, San-Cang fault, Mu-Ru fault, and Peng-Qi fault are the second-order faults in the region. They are the main ore-controlling structures of gold deposits.

The Dongfeng Mining area, Jiuqu Sub-Mine of Linglong Gold Mine, Fushan Gold Mine, Lingnan Gold Mine, Luoshan Gold Mine, Damoqujia Gold Mine, and the Dayingezhuang, Xiadian, and Jiangjiayao gold deposits are located in the Zhaoping-Ping fault zone. The Hexi mining area, Hedong mining area, Shangzhuang mining area, and Jinhuashan mining area are located in the Long-Lai fault zone. The Xishan sub-deposit of the Sanshandao gold mine is located in the San-Cang fault zone. The Jinqingding mining area and Shuangshan-tun mining area are located in the Mu-Ru fault zone. The Yanshan mining area, Yankou mining area, and Menlou mining area are located in the Peng-Qi fault zone.

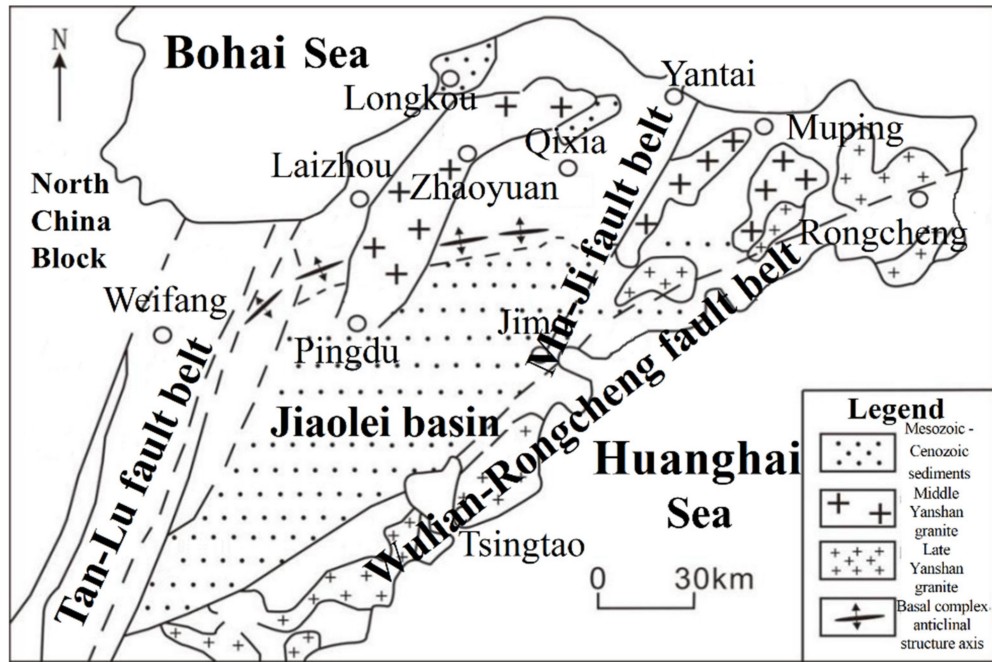

**Figure 4.** Brief map of tectonic geology in Jiaodong area.

*2.2. Production Situation*

Among the nineteen mines, there are nine large mines with an annual production scale of more than 150,000 tons, nine medium-sized mines with an annual production capacity of 60,000–150,000 tons, and one small mine with an annual production capacity of fewer than 60,000 tons. These mines have a total designed annual production capacity of 7.9301 million tons.

Nine mines adopt the combined development method of the open shaft and blind shaft. Two mines adopt the combined development method of the open shaft and (auxiliary) ramp. One mine adopts the combined development method of the open shaft, inclined shaft, and blind shaft. One mine adopts the combined development method of drift, open shaft, blind shaft, and blind inclined shaft. There is one mine with a development depth of less than 800 m, twelve mines with depths of 800~1000 m, and six mines with depths of more than 1000 m. The mine with the deepest development depth is the Zhaojin Xiadian gold mine, which is 1565 m. There are eleven mines with the main production horizon above −800 m and one mine with −800~1000 m, and the rest of the mines are in a state of shutdown or have not yet been put into production.

## 3. Present Situation of Deep Ground Pressure in Non-coal Mines

*3.1. Regional Tectonic Stress Level*

Related studies have calculated the linear regression of the relationship between the maximum horizontal principal stress and the minimum horizontal principal stress with depth in the northwest of Jiaodong. The results show that when the depth reaches 1000 m, the maximum horizontal principal stress is about 40 MPa. Most of the levels of minimum horizontal principal stress are close to 20 MPa, and the highest can reach 30 MPa, which belongs to the high ground stress areas.

In the Sanshandao Xiling mining area, the coring hole position is determined according to a comprehensive and significant principle. The coring line is arranged in a cross shape with the 88# borehole exploration line as the axis, taking the northeast and southwest directions into account. Finally, five representative prospecting boreholes are identified, as shown in Figures 5 and 6.

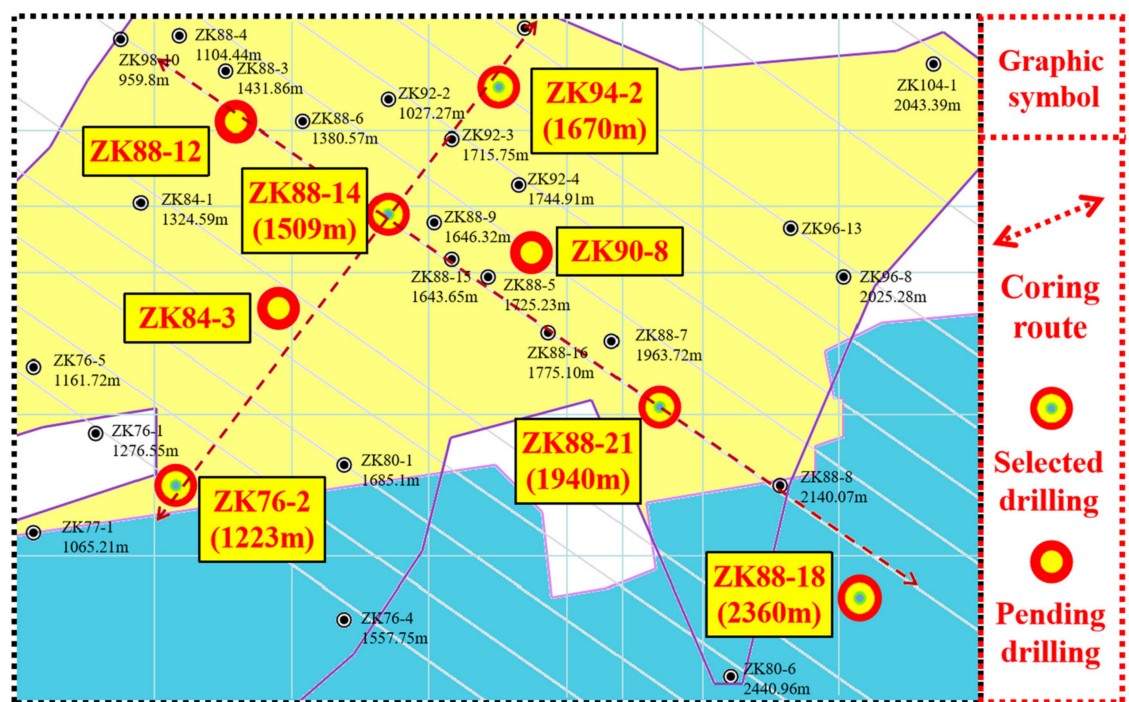

**Figure 5.** Selection of core drilling location.

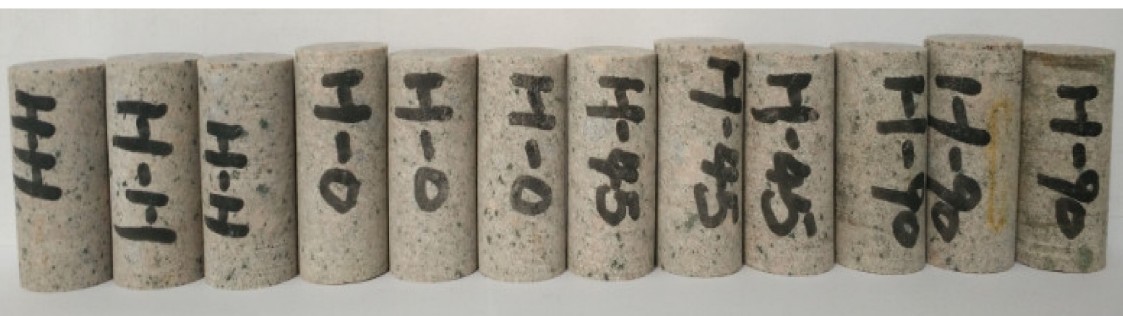

**Figure 6.** Part of the experimental specimens.

In the Xincheng Gold Mine, based on core reorientation technology [34], the in situ stress measurement is carried out by drilling cores with seven different depths of −950 m, −1050 m, −1150 m, −1250 m, −1350 m, −1450 m, and −1550 m. The in situ stress is measured by acoustic emission. The results are shown in Table 2.

**Table 2.** Stress distribution at different depths, ref. [35].

| Measuring Point Depth (m) | Vertical Principal Stress $\sigma_v$ (MPa) | Self-Weight Stress $\sigma$ (MPa) | Maximum Horizontal Principal Stress $\sigma_H$ (MPa) | Minimum Horizontal Principal Stress $\sigma_h$ (MPa) | Horizontal Maximum Principal Stress Direction θ |
|---|---|---|---|---|---|
| 950 | 26.41 | 26.46 | 32.78 | 14.52 | N103°34′ E |
| 1050 | 29.22 | 29.16 | 41.83 | 17.18 | N103°42′ E |
| 1150 | 33.33 | 31.86 | 35.37 | 16.64 | N95°43′ E |
| 1250 | 34.25 | 34.56 | 38.95 | 16.85 | N110°13′ E |
| 1350 | 35.98 | 37.26 | 41.23 | 16.44 | N105°26′ E |
| 1450 | 40.01 | 39.96 | 40.63 | 14.02 | N111°20′ E |
| 1550 | 40.95 | 42.66 | 45.84 | 17.56 | N107°33′ E |

It is shown that the vertical principal stress, the self-weight stress, and the minimum horizontal principal stress increase the same amount at the depth of 1050 m, which is about 3 MPa. When the depth increases to 1350 m, the vertical principal stress increases by about

3% per 100 m, and the self-weight stress and the maximum horizontal principal stress maintain a steady growth rate of about 3 MPa per 100 m. When the depth increases from 1350 m to 1550 m, due to the influence of geological structure, the vertical principal stress increases rapidly by 14%, while the self-weight stress continues to increase at a relatively stable rate. It is worth noting that due to the existence of geological structures, both the maximum horizontal principal stress and the minimum horizontal principal stress decrease slightly from depths of 1350 m to 1450 m, and then rise slightly to the normal level at the speed of 16% with the increase in depth.

(1) The principal vertical stress increases nearly linearly with the increase in coring depth, which is consistent with the buried depth's dead weight stress (Figure 7).

(2) The in situ stress in the borehole is located mainly in horizontal tectonic stress. In the different depths of the boreholes, the maximum stress is the horizontal maximum principal stress, and the intermediate principal stress is the vertical stress. With the increase in the borehole depth, the dominant role of the horizontal tectonic stress field weakens, and the effect of the self-weight stress field tends to increase, as shown in Figure 8.

(3) The horizontal maximum principal stress increases with the increase in hole depth. It is worth noting that the maximum principal stress shows a sudden increase at −1050 m, reaching 41.83 MPa (higher than the stress value of −1150 m and −1250 m at greater depths). This indicates an unstable local change in the maximum principal stress of this area.

(4) The azimuth of the maximum horizontal principal stress at different depths is consistent; all values are along the NWW~SEE direction, between N95° E and N111° E, as shown in Figure 9. This shows that the location of the new shaft of the Xincheng Gold Mine is consistent with its geological structure and the orientation of the maximum principal stress measured by many scientific institutions above its −950 m elevation.

(5) The maximum horizontal principal stress ratio to the principal vertical stress (lateral pressure coefficient) in the measured area is between 1.01 and 1.43, which is consistent with the distribution law of the lateral pressure coefficient in China [36,37].

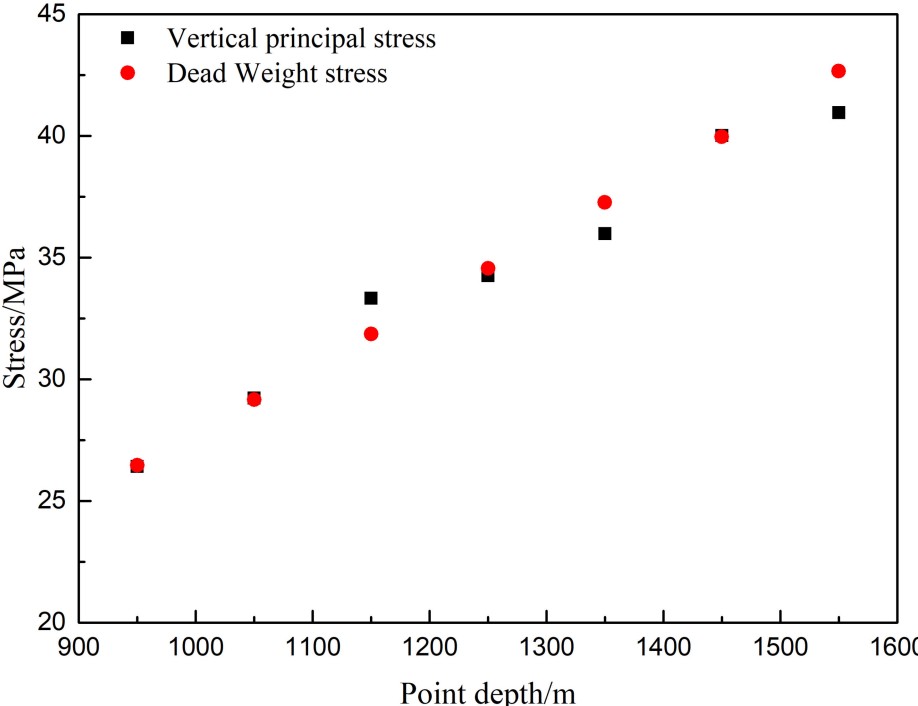

**Figure 7.** Relationship between in situ vertical stress, self-weight stress, and drilling depth.

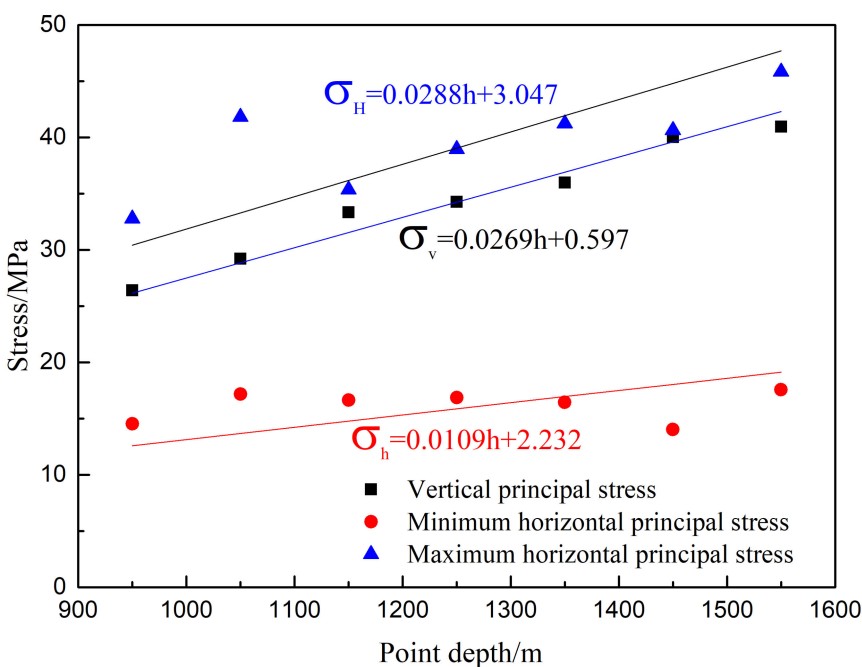

**Figure 8.** Comparison of stresses in different directions.

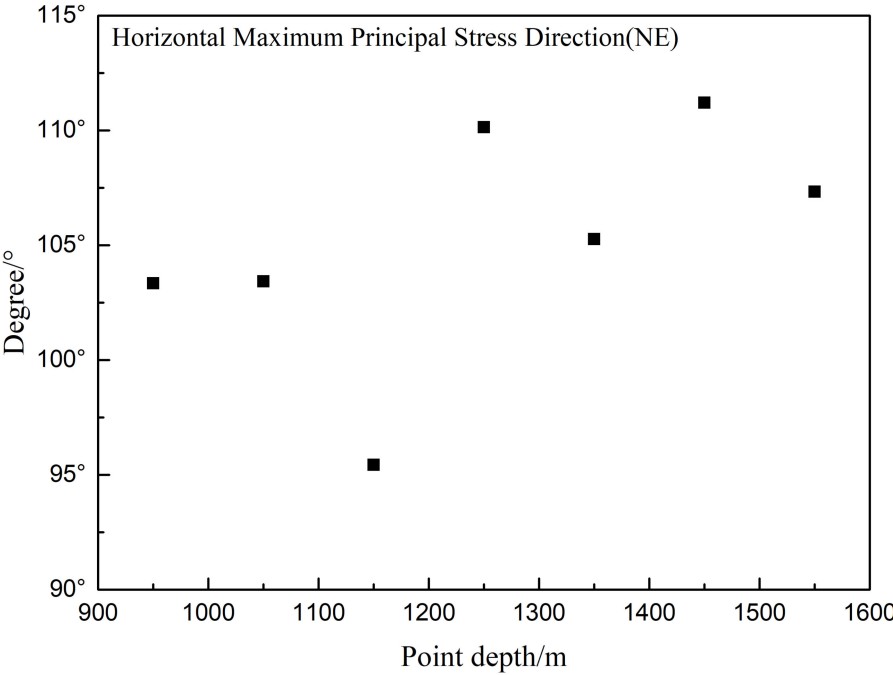

**Figure 9.** Variation trend of horizontal maximum principal stress direction with depth.

### 3.2. Deep Ground Pressure Appearance

In the roadways and stope mines with mining depths of more than 800 m, rock mass instability has occurred to varying degrees in some mines, and slight rockburst has occurred in a few mines. In the deep development process, some mines have a failure tendency of roadway surrounding rock. This is because of the complex regional geological structure, broken rock mass, high in situ stress level, or because the roadway axis is not parallel to the direction of horizontal maximum principal stress. Few mines have long mining histories and many pressure concentration areas in the lower part of the goaf have formed at early mining stages. Therefore, slight rockburst phenomena such as rock mass ejection have occurred.

Most of the investigated areas of production and construction in the mines are stable, and there is no obvious ground pressure. There are also slight roof falls, slopes, or rock spalling in some local areas of the mines, as shown in Figure 10.

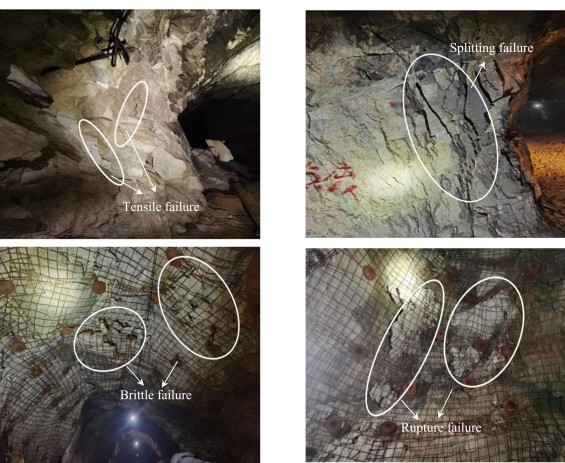

**Figure 10.** Deep ground pressure appearance.

### 3.3. Control Measures of Deep Ground Pressure

At present, bolting and shotcreting net flexible support measures are widely used to control rock mass instability in deep non-coal mines in Shandong province. The Jinqingding mining area of the Jinzhou Mining Group is gradually replacing rigid support with flexible support and uses advanced grouting in local broken areas [38]. In the Sanshandao gold mine, resin bolts are widely used in deep support to replace pipe joint bolts, anchor-shotcreting nets, and piercing belts for support when excavating. At the same time, the Jinzhou Mining Group actively cooperates with relevant departments to develop prestressed anchors and constantly optimize its deep support technology [39–41]. Long anchor cables are used to support the stope's roof in the Jiangjiayao gold mine.

### 3.4. Risk of Rockburst in Deep Mines

Among the nineteen mines investigated, the typical lithology of the upper and lower surrounding rock was amphibolite, monzonitic granite, sericite, granite, potash granite, granulite, and so on. Therefore, the evaluation result of the rockburst tendency based on the drilling core of the Shanjin Sanshandao gold mine is instructive in evaluating the rockburst tendency in the region. Typical monzonitic granite, granite, and sericite have more than moderate rockburst tendencies (Figure 11). With the increase in burial depth, the corresponding intensity grade increases.

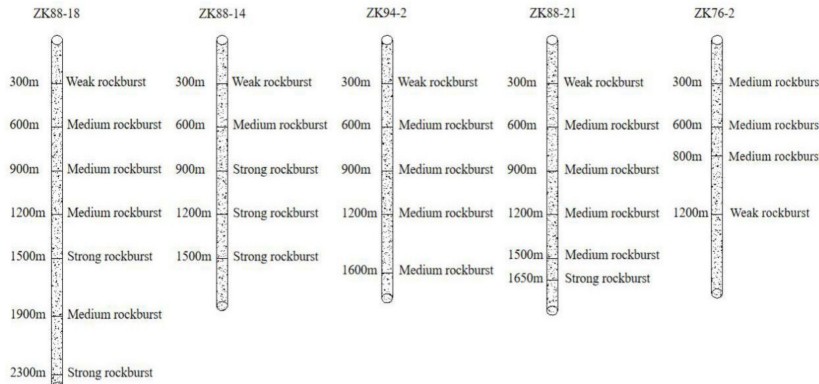

**Figure 11.** Schematic diagram of rockburst tendency grade of boreholes varying with depth in Xiling mining area of Sanshandao.

## 4. Problems Existing in the Mines

### 4.1. The Deep Development System Is Complex

Overall, the deep development system of the investigated mines is more complex, and the development engineering includes open shafts, blind shafts, blind inclined shafts, ramps, and so on. Commonly, there are more than ten shafts in a mine. From the point of view of mine construction, the mining projects are not reasonably planned. Design and construction are often needed due to later expansion and capacity expansion, resulting in the complexity of the development system. As a result, there is a high cost for mine capital construction. Additionally, system maintenance requires a heavy workload. Furthermore, it is difficult to manage multiple lifting systems and there are many hidden dangers in the processes of operation.

### 4.2. Lack of Rock Mechanics Data in Most Deep Mines

Among the nineteen mines, the Shanjin Linglong Jiuqu mine, Shanjin Dongfeng mining area, Zhaojin Xiadian gold mine, Zhaojin Dayingezhuang gold mine, and Zhongkuang Lingnan No. 5 mining area have carried out preliminary research on deep rock mechanics and accumulated some basic data on rock mechanics. The Sanshandao gold mine has carried out more systematic rock mechanics research, while the other thirteen mines have not yet carried out rock mechanics data collection or related research. The research on in situ stress measurement, rock mass quality classification, and rock mass stability analysis is an essential reference for mining engineering design.

Mine rock mechanics data are the basis of mining design. With the increase in mining depth, engineering designs based on empirical analogies gradually show some problems of unsuitability and low reliability. Two problems have been found in the investigation of this paper. First, most of those working in the mines lack an understanding of the importance of rock mechanics data collection, management, and ideological research work. Second, the obtained basic data of rock mechanics have locality and discontinuity. Usually there are only simple strength test data or partial area data, which lack systematic studies, and the guiding significance for mining engineering design is limited.

### 4.3. Some Support Engineering in Deep Mining Is Unsuitable and Not in Place

Nineteen mines have adopted the necessary shotcreting or combined anchor supports, shotcreting nets, or steel arch supports in unstable areas, especially in areas with broken structures. The roofs of some unstable areas have adopted long anchor cable supports. However, there are still three problems.

First, the support scheme in some areas is unsuitable. For example, roadways with obvious deformations adopt rigid support under high stress. This often does not have an excellent effect on support, but it increases the cost of the support due to the deformation and failure of the support.

Second, the support is not timely nor in place. For example, when the stope, roadway, and drift are temporarily stable in the deep part, many mines do not take or take insufficient support measures to save support costs.

Third, the support design is unsuitable due to the lack of basic rock mechanics data and rock mass stability classification standards. Moreover, the selection of support parameters lacks references, resulting in excessive support or support space.

### 4.4. The Ground Pressure Monitoring System Lacks Maintenance, and the Early Warning Ability Is Poor

Among the investigated nineteen mines, thirteen mines have installed ground pressure online monitoring systems, and some mines have also installed several monitoring systems. However, only a few of the mines with the system can operate normally, and the monitoring and early warning systems are poor, so there are the following problems:

① After installation, most online ground pressure monitoring systems are not inspected and maintained. A few mines carry out maintenance after equipment damage,

which lacks timeliness, resulting in the loss of a lot of monitoring data and the omission of microseismic events.

② The monitoring data cannot be analyzed and processed in time. Only a few mines entrust technical service institutions for data analysis. There is a severe lag in data analysis, and the ground pressure online systems' monitoring and early warning functions cannot be fully utilized.

③ Transmission formats of monitoring data are various. The online monitoring systems of ground pressure in mines investigated are independently developed by domestic manufacturers, universities, or scientific research institutes, and some are imported from abroad. As a result, the data storage formats are various. The intellectual property rights of imported equipment are strictly protected abroad, so there are some difficulties converting different data formats into a unified standard format, which causes some obstacles to the integrated management of the online monitoring system of underground non-coal mine ground pressure.

④ The ground pressure online monitoring system has a low positioning accuracy and a poor early warning ability for microseismic events. The online microseismic monitoring system of ground pressure has difficulty locating rock fractures accurately. At the same time, the installed online monitoring system of ground pressure only has a real-time monitoring function and cannot reduce noise in real time. The risk grade of ground pressure disasters cannot be judged, so it is challenging to implement ground pressure disaster prediction and early warning.

### *4.5. The Temperature in the Deep Mine Is High, the Humidity Exceeds the Standard, and the Ventilation System Is Not Perfect*

At present, some non-coal mines with a mining depth of over 800 m are suffering from heat damage caused by surrounding rocks and groundwater during mining [40]. Imperfect ventilation systems and lagging ventilation projects result in high local working surface temperatures, excessive humidity, and a poor working environment, which affects the efficiency of work and equipment.

### *4.6. The Intelligence Level of Deep Mine Is Poor*

Except for the Sanshandao gold mine, the intelligence levels of the other mines are relatively poor. Whether it is the construction of the related intelligent data platforms, centralized management, the control platform, underground mining, excavation, loading, or transportation equipment, there is a significant gap in the implementation of intelligent mining.

## 5. Prevention and Control Countermeasures and Suggestions

Given the above problems and the current situation of each production system, the following suggestions are put forward.

### *5.1. Establish and Improve the Working Organization and System of Safety Risk Prevention in Deep Mining*

To prevent the safety risks, deep mines should establish and improve the work organization and safety risk prevention system, and set up a leading group for safety risk prevention headed by technical personnel and management personnel.

### *5.2. Comprehensively Consider the Planning and Layout and Optimize the System Layout as a Whole*

In mine construction planning, all deep mines should consider the overall layout and optimize the system layout to avoid a large number of new development projects in the process of production succession.

### 5.3. Start the Work of Mine Rock Mechanics and Establish the Working System of Rock Mechanics

To ensure the safety of deep mining, all deep mines should be equipped with engineers and technicians to collect rock mechanics data, such as investigations on rock mass structural planes, rock sampling, physical and mechanical index tests, in situ stress tests, point load index tests, and rock mass quality classification and stability analyses, etc. A computer-aided system for deep rock mass stability and support should be developed to provide guidance for the deep shaft and roadway, the stope design, and the support design and to improve the rationality and reliability of the design.

### 5.4. Carry Out Ground Pressure Investigation

It is necessary to investigate the ground pressure of the roadway, stope, and drift with a mining depth of more than 800 m, record the instability and failures of rock mass and support failures, and form investigation files. Ground pressure monitoring should be strengthened for the areas with frequent ground strata behaviors.

### 5.5. Test and Evaluate the Tendency of Rockburst

Deep mines should entrust scientific institutions to conduct rockburst tendency tests and evaluate the possible rockburst intensity by prospecting borehole cores or sampling in deep mining faces. Pressure relief or energy absorption support methods should be adopted in moderate rockburst areas.

### 5.6. Strengthen the Maintenance and Management of the Online Monitoring System of Ground Pressure

All deep mines should be equipped with maintenance and management personnel for the online ground pressure monitoring system to timely process monitoring data, identify microseismic events, compile weekly and monthly monitoring data reports, statistically analyze the law of ground pressure activities, and provide early warning information. Thereby maintenance and inspections can ensure the normal operation of the monitoring system.

### 5.7. Establishment of a Unified Ground Pressure Monitoring and Early Warning Platform

To fully implement the monitoring and early warning function of the ground pressure monitoring system, it is necessary to unify the online monitoring and early warning platform for underground non-coal mining pressure. This can help implement the unified management and integrated display of ground pressure monitoring data. Mines should carry out relevant positioning algorithms and early warning indicators and models to improve ground pressure monitoring, early warning, and supervision.

### 5.8. Innovate the Concept of Support Methods

Deep mines should establish a computer-aided decision-making system for deep support based on fundamental data, such as the rock's physical and mechanical parameters, in situ stress, joints and fissures, rock quality index, and groundwater state, to implement an intelligent dynamic stability classification of ore and rock. Mine companies should determine a reasonable and practical deep support system. Under deep high stress, the support system should specifically address energy absorption and deformation.

### 5.9. Strengthen Cooling Measures

Deep mines should optimize the design of deep ventilation systems to meet the ventilation needs under severe conditions, expedite the construction of deep ventilation systems, build complete deep ventilation systems in time, and strengthen the operation and management of deep ventilation systems.

### 5.10. Improve the Intelligent Level of Deep Mining

Deep mines should continuously strengthen scientific and technological investment, carry out digital and information system construction, and centralize the management and control

of various production systems, such as backfilling, lifting, drainage, and ventilation. Mines can popularize and apply advanced large-scale mining, excavation, loading, and transportation equipment and make full use of big data, cloud computing, the Internet of things, 5G communication, and other technologies to improve the intellectual level of deep mining.

## 6. Conclusions

This manuscript studies the current situation of main safety systems in non-coal mines with a mining depth of more than 800 m in Shandong province and mainly draws the following conclusions:

(1) There are no large-scale ground pressure behaviors in underground non-coal mines with a mining depth of more than 800 m in Shandong province. No destructive ground pressure accidents have occured, indicating they are still in the early stages of ground pressure disasters, and rock mass instability can be controlled by strengthening support. However, with the increase in mining depth, the rockburst risk rises, and the support work becomes difficult, so it is necessary to strengthen scientific study and explore effective control measures.

(2) All the mines with normal production and construction have built ground pressure online monitoring systems. However, few systems can play an effective role in monitoring and early warning. Hence, it is necessary to strengthen the maintenance and management of the ground pressure monitoring system and establish a unified ground pressure monitoring and early warning platform for the whole province. The safety facilities of each mine meet the needs of safe production. However, their maintenance and management in the operation process need to be further strengthened, and the level of intelligence needs to be improved.

(3) The phenomena of high temperatures and high humidity in the deep mines are relatively common, and cooling can be achieved by strengthening ventilation at the current mining depth. Each mine needs to improve its ventilation systems further, strengthen ventilation management, and adopt local cooling if necessary.

(4) The construction of safety facilities in a few of the surveyed gold mines is similar to that of international mines with high development levels, but other mines can only meet the minimum requirements. It is necessary to introduce advanced digital intelligent mine technology, such as ground pressure monitoring and early warning systems, as well as big data platforms for upgrading and transforming mine construction in the future.

**Author Contributions:** Formal analysis, L.C.; Funding acquisition, X.L. (Xingquan Liu); Investigation, Q.W.; Methodology, C.W. and J.M.; Project administration, K.C.; Resources, X.L. (Xuelong Li); Software, H.L. All authors have read and agreed to the published version of the manuscript.

**Funding:** This work was financially supported by the National Natural Science Foundation of China (52104204, 52204226), the Natural Science Foundation of Shandong Province (ZR2021QE170), Qingdao Postdoctoral Applied Research Project, and China Postdoctoral Science Foundation Funded Project (2022M711961).

**Data Availability Statement:** The data that support the findings of this study are available from the corresponding author upon reasonable request.

**Conflicts of Interest:** The authors declare that there are no conflicts of interest regarding the publication of this paper.

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
