# Peer review of "Safety and Protection Measures of Underground Non-Coal Mines with Mining Depth over 800 m: A Case Study in Shandong, China"

_sustainability, doi:10.3390/su142013345_

Round 1

Reviewer 1 Report

It is fairly established that the deep mining brings forth a plethora of challenges, especially to the safety, health, environment and overall productivity of the system. Also, it is very well known that these challenges make the deep mines vulnerable to hazards, risk, largely due to insurmountable rock stress changes and behaviour. Some of the Gold mines in Africa have demonstrated it. 

The present paper does not bring forth some new and novel findings in the case study mines.

Development of support system, ground measurement system, temperature, gas pressure and humidity etc  by use of wireless network communication , information technology, AI and big data analytics for prognostication and remediation of the challenges need to be addressed aptly to enhance the knowledge and incite interest of the readers. The paper, in its present form, is not only lengthy but beats the old drum. 

It needs significant improvements before considering for publication.

Author Response

Dear editors and reviewers,

I am very glad to receive your comments. Thanks for your comments concerning our manuscript named “Safety Situation and Protection Measures of Underground Non-coal Mines with Mining Depth over 800 m: A Case Study in Shandong, China”. Those comments are all valuable and very helpful for us to revise and improve our manuscript. The comments are also of significant importance to guide our research. After studying the comments carefully, we have made the modifications and we hope they can meet your approval and requirements. The main corrections and responses are as follows.

We hope with these revisions, our manuscript can be accepted to publish in the Sustainability.

Looking forward to hearing from you.

Best regards

Reviewer #1’s comments:

It is fairly established that the deep mining brings forth a plethora of challenges, especially to the safety, health, environment and overall productivity of the system. Also, it is very well known that these challenges make the deep mines vulnerable to hazards, risk, largely due to insurmountable rock stress changes and behaviour. Some of the Gold mines in Africa have demonstrated it.

The present paper does not bring forth some new and novel findings in the case study mines.

Development of support system, ground measurement system, temperature, gas pressure and humidity etc by use of wireless network communication, information technology, AI and big data analytics for prognostication and remediation of the challenges need to be addressed aptly to enhance the knowledge and incite interest of the readers. The paper, in its present form, is not only lengthy but beats the old drum.

It needs significant improvements before considering for publication.

Response:

Thank you so much for these constructive comments.

In fact, this paper focuses on the promotion and application of wireless network communication, artificial intelligence, big data and other advanced technologies in deep mining when investigating the mine safety situation. Regrettably, these advanced technologies have been tried in very few mines for a variety of reasons. After investigation, we believed that this is related to the economic efficiency and reliability of the technology. Therefore, we give corresponding suggestions in the measures section, hoping that these suggestions can be recognized and promoted in the future.

In addition, we have noticed that there are duplicate expressions in the paper and have modified these statements.

It is hoped that these improvements will enable the article to meet the requirements of publication.

Reviewer 2 Report

In the paper “Safety Situation and Protection Measures of Underground Non-coal Mines with Mining Depth over 800 m: A Case Study in Shandong, China”, a typical case has been selected to study. The following are the comments for the paper. Generally, typical cases are instructive, but details need improvement.

(1) Please check the correct expressions of some professional terms, for example, the mountain pressure disaster.

(2) The description in Table 1 is not concise enough. The description of the ownership of the mine seems not necessary. Please explain the reasons for this or adjust it.

(3) Pay attention on the spelling of proper nouns in Figure 4. Please correct similar problems in the manuscript.

(4) Authors should enhance the description of the manuscript's conclusion to make it more informative.

(5) The express requires improvement, and it is recommended that the authors obtain an improvement.

Author Response

Reviewer #2’s comments:

In the paper “Safety Situation and Protection Measures of Underground Non-coal Mines with Mining Depth over 800 m: A Case Study in Shandong, China”, a typical case has been selected to study. The following are the comments for the paper. Generally, typical cases are instructive, but details need improvement.

(1) Please check the correct expressions of some professional terms, for example, the mountain pressure disaster.

Response:

Thank you so much for this comment.

We find that the expressions of some professional terms in the article are not appropriate. We corrected them after extensive literature review, for example, we corrected the mountain pressure disaster to ground pressure disaster. We hope these changes will make the article more rigorous.

(2) The description in Table 1 is not concise enough. The description of the ownership of the mine seems not necessary. Please explain the reasons for this or adjust it.

Response:

Thanks for your constructive suggestion.

To make the chart more concise, we use abbreviations for ownership and production status. In addition, we consider the description of Table 1 is necessary. First of all, the content of Table 1 is compared with Figure 1, and the information of the mine is supplemented. Secondly, mines with different ownership also have different production plans and construction plans. It is necessary to explain these differences as part of the mine profile in detail. In order to show these differences comprehensively, we chose the form of table.

(3) Pay attention on the spelling of proper nouns in Figure 4. Please correct similar problems in the manuscript.

Response:

Thanks for your reminder.

I'm sorry for the problem in our manuscript. We have corrected the errors and checked other figures. We hope these works can make the article more rigorous.

(4) Authors should enhance the description of the manuscript's conclusion to make it more informative.

Response:

Thank you so much for this constructive comment.

We find that some expression in the abstract and conclusions are not clearly, so we added and modified these parts. We hope that these modifications can make the description of the research results clearer and more credible.

(5) The express requires improvement, and it is recommended that the authors obtain an improvement.

Response:

Thank you so much for this constructive comment.

We find that some of the sentences in this paper are not rigorous enough or do not conform to the norms. We modified these expression and sentences, checked and modified the format of the article according to the standards of international journals. We hope these works can improve the quality of the article to meet the requirements.

Reviewer 3 Report

This paper investigates the appearance of mining pressure, ground pressure monitoring and the operation of “six systems” of non-coal mine with a mining depth of more than 800m in Jiaodong area, China. The topic of the paper is clear and the results are interesting, while there is still some question should be addressed before the paper could be accepted:

1. In Introduction, a further detailed review of relative literatures is needed to illustrate the novelty of this study. It is suggested that some new references should be cited.

2. The introduction and conclusions should be refined to highlight the focus and novelty of the study.

3. Please carefully refer to the published articles in this journal to check the format of figures, charts and paragraphs.

4. Please check the format of the citations and references and modify them according to journal requirements.

5. What are the implications of this study for deep gold mining?

6. According to the author's experience, what technology should be improved in Chinese gold mining compared with other gold mining countries in the world?

Author Response

This paper investigates the appearance of mining pressure, ground pressure monitoring and the operation of “six systems” of non-coal mine with a mining depth of more than 800m in Jiaodong area, China. The topic of the paper is clear and the results are interesting, while there is still some question should be addressed before the paper could be accepted:

  1. In Introduction, a further detailed review of relative literatures is needed to illustrate the novelty of this study. It is suggested that some new references should be cited.

Response:

Thank you so much for this constructive comment and suggestion.

We have taken note of your suggestions for the cited references. We have added some newly published high-quality literature, made an in-depth study of their researches and compared it with the work of this paper. Citing high-quality literature can describe the current research progress more clearly and enable readers to have a more comprehensive and in-depth understanding of the article.

  1. The introduction and conclusions should be refined to highlight the focus and novelty of the study.

Response:

Thank you so much for this constructive comment.

We find that some expression in the introduction and conclusions are not refined, so we modified these parts. We hope that these modifications can make the description of the research results clearer and concise.

  1. Please carefully refer to the published articles in this journal to check the format of figures, charts and paragraphs.

Response:

Thanks for your reminding.

We feel sorry that the format of some figures and tables in our manuscript does not meet the requirements of the journal. We have checked and modified these contents. We hope that these modifications can make the article meet the published standards.

  1. Please check the format of the citations and references and modify them according to journal requirements.

Response:

Thank you very much for your reminding.

We feel sorry that we have not been able to type the references in accordance with the requirements of the periodical, but we have now adjusted the content and format of the literature to meet the requirements for publication in your journal.

  1. What are the implications of this study for deep gold mining?

Response:

Thank you so much for this constructive comment.

This paper conducts a safety survey of some gold mines in Shandong Province, China, focusing on mine production safety and related facilities, and comprehensively compares and describes each mine. In addition, we selected the most representative Xiling mining area in Sanshandao, northwestern Jiaozhou, and selected the drilling site according to the principles determined in the previous study. The extracted cores were tested to determine the regional tectonic stress level. Section 5 of the article gives suggestions for prevention and control, and also contains the enlightenment obtained after comprehensive comparative analysis. We add summary and enlightenment in the conclusion part. We hope these modifications can make the article more comprehensive.

  1. According to the author's experience, what technology should be improved in Chinese gold mining compared with other gold mining countries in the world?

Response:

Thank you so much for this constructive suggestion.

In the investigation of mine safety, we compared domestic mines with internationally developed mines, found and pointed out the technologies that need to be improved. We add improvement suggestions to the conclusion. We hope that these modifications can make the article more comprehensive and rigorous.

Reviewer 4 Report

Journal: Sustainability (ISSN 2071-1050)

Manuscript ID: sustainability-1923084

Type: Article

Number of Pages: 18 

Title

Safety Situation and Protection Measures of Underground Non-coal Mines with Mining Depth over 800 m: A Case Study in Shandong, China

Authors:

Li Cheng, Qinzheng Wu, Haotian Li, Kexu Chen, Chunlong Wang, Xingquan Liu, Xuelong Li and Jingjing Meng

Review report

The article depicts ground pressure hazard caused by increasing depth (from 800 to 1350 meters) of exploitation in 19 yellow gold mines situated in the largest gold-producing area in China. Measuring the in-situ stress can help define accurate prevention and control of ground pressure disasters. At present, in-situ stress surveys are carried out in numerous countries but in China, they are insufficiently applied. The main goal of detecting underground stress in deep mines is to determine the seismic activity and mode of ground stress, the direction of main stress, and its changes with depth. It is suggested that the corresponding stress and seismic intensity will increase with mining depth. The increasing ground pressure may have an impact on production safety, so some mines adopted new support schemes for instance they gradually replace rigid support with flexible support and advanced grouting is used in local broken areas. In the investigated gold mine, the resin bolt is widely used in deep support to replace the pipe joint bolt, anchor-shotcreting net, and piercing belt to support during excavation. These measures effectively reduced the damage caused by ground pressure, but the high cost made it difficult to be implemented in all mines. The basic situation in 19 gold mines was described as well as their production and deposit geological settings. The present situation of deep ground pressure is presented. Regional tectonic stress levels and stress distribution at different depths are depicted using the data from boreholes (vertical principal stress, self-weight stress, maximum horizontal principal stress, minimum horizontal principal stress, horizontal maximum principal. The pressure measured in boreholes was analyzed. The in-situ stress in the borehole is located mainly in horizontal tectonic stress. At the different depths of boreholes, the maximum stress is the horizontal maximum principal stress, and the intermediate principal stress is vertical stress. With the increase of the borehole depth, the dominant role of the horizontal tectonic stress field weakens, and the effect of the self-weight stress increases. The horizontal maximum principal stress increases with the increase of depth. The maximum principal stress shows a sudden increase at 1050  m, reaching 41.83 MPa (higher than the stress value of -1150 m and -1250 m at greater depth). This indicates an unstable local change in the maximum principal stress in this area. The azimuth of the maximum horizontal principal stress at different depths is consistent, all along the NWW~SEE direction, between N95 °E and N111 °E. This shows that the location of the new shaft of one Gold Mine is consistent with its geological structure and the orientation of the maximum principal stress measured by many scientific institutions above its -950 m elevation. The maximum horizontal principal stress ratio to the principal vertical stress (lateral pressure coefficient) in the area is between 1.01 and 1.43, which is consistent with the distribution law of the lateral pressure coefficient in China. The test results show that at a depth bigger than 1050 m, the increase of vertical principal stress is equal to that of minimum horizontal principal stress, i.e. about 3 MPa per 100 m. When the depth achieves 1350 m, the vertical principal stress increases by about 3 % per 100 m, and the self-weight stress and the maximum horizontal principal stress grow steadily at about 3 MPa per 100 m. Moreover, the test results make it possible to check the ground pressure monitoring system in each mine. The deep ground pressure appearance was depicted too as well as control measures of deep ground pressure including supporting. The investigation results show that in the roadway and stope situated deeper than 800 m rock mass instability occurred in some mines, and in about 5% of mines low energy rock bursts take place, There was a high-stress area in the lower part of the old goaf, while the area of normal production and construction was stable, without an excessive ground pressure. The risk of rock bursts in deep mines is shortly depicted using a schematic diagram of the rock burst tendency grade of boreholes varying with depth. The evaluation result of rock burst tendency based on the drilling core of the investigated mine is instructive in evaluating rock burst tendency in the region. The typical monzonitic granite, granite, and sericite have more than moderate rock burst tendencies. With the increase of burial depth, the corresponding intensity grade of rock bursts will increase. The authors present problems existing in the mines such as 1) The deep development system is complex, 2) Lack of rock mechanics data in most deep mines, and 3) Some support engineering in deep mining is unreasonable and not in place, 4) The ground pressure monitoring system lacks maintenance, and the early warning ability is poor, 5) The temperature in the deep mine is high, the humidity exceeds the standard, and the ventilation system is not perfect, 6) The intelligence level of the deep mine is low.  The authors suggest taking certain activities to reduce, monitor and prevent the safety risk. 

The article is divided into 6 sections and 6 subsections: 1. Introduction; 2. Basic Situation of Mines: 2.1. Geological Condition, 2.2. Production Situation; 3. Present Situation of Deep Ground Pressure in Non-coal Mines: 3.1. Regional Tectonic Stress Level, 3.2. Deep Ground Pressure Appearance Sample and parameters selection, 3.3. Control Measures of Deep Ground Pressure, 3.4. Risk of Rockburst in Deep Mines; 4. Problems Existing in the Mines; 5. Prevention and Control Countermeasures and Suggestions; 6. Conclusion. There are 11 figures and 2 tables in the article.

Areas of the study strength

The Authors performed analyses and try to fill part of the gap in the information about the monitoring of ground pressure and rock burst hazard in deep gold mines in China. The most important problems in deep mines were noted and shortly depicted. Some measures to improve the situation were presented.

Areas of the study weakness

The article is not scientific enough. There are a lot of depictions, suggestions, and tips which do not make the article scientific. The Authors should keep the introduction comprehensible to scientists working outside the topic of the paper. The conclusion is not good enough. It should be corrected and include deeper insight into the results and their usability. The materials and methods were not depicted. There is no information about tests made on the cores. How were they conducted? How many tests? What was measured? What are the regressions calculated for? What about the rock burst hazard? How the tendency was assessed?

There is a kind of chaos throughout the whole article, which needs to draw the Authors’ attention. The article was written without enough care, which should be corrected.

The design of the article is not clear and logical enough. More consistency would be appreciated and would improve the article’s quality.

Moreover, English should be improved. There are a lot of grammar errors for example inconsistency between an agent and a verb, the syntax is also not good enough, word order and choice are poor as well as a lot of wordiness.  The best way to improve English is by a native speaker consulting

Author Response

Reviewer #4’s comments:

The article depicts ground pressure hazard caused by increasing depth (from 800 to 1350 meters) of exploitation in 19 yellow gold mines situated in the largest gold-producing area in China. Measuring the in-situ stress can help define accurate prevention and control of ground pressure disasters. At present, in-situ stress surveys are carried out in numerous countries but in China, they are insufficiently applied. The main goal of detecting underground stress in deep mines is to determine the seismic activity and mode of ground stress, the direction of main stress, and its changes with depth. It is suggested that the corresponding stress and seismic intensity will increase with mining depth. The increasing ground pressure may have an impact on production safety, so some mines adopted new support schemes for instance they gradually replace rigid support with flexible support and advanced grouting is used in local broken areas. In the investigated gold mine, the resin bolt is widely used in deep support to replace the pipe joint bolt, anchor-shotcreting net, and piercing belt to support during excavation. These measures effectively reduced the damage caused by ground pressure, but the high cost made it difficult to be implemented in all mines. The basic situation in 19 gold mines was described as well as their production and deposit geological settings. The present situation of deep ground pressure is presented. Regional tectonic stress levels and stress distribution at different depths are depicted using the data from boreholes (vertical principal stress, self-weight stress, maximum horizontal principal stress, minimum horizontal principal stress, horizontal maximum principal. The pressure measured in boreholes was analyzed. The in-situ stress in the borehole is located mainly in horizontal tectonic stress. At the different depths of boreholes, the maximum stress is the horizontal maximum principal stress, and the intermediate principal stress is vertical stress. With the increase of the borehole depth, the dominant role of the horizontal tectonic stress field weakens, and the effect of the self-weight stress increases. The horizontal maximum principal stress increases with the increase of depth. The maximum principal stress shows a sudden increase at 1050 m, reaching 41.83 MPa (higher than the stress value of -1150 m and -1250 m at greater depth). This indicates an unstable local change in the maximum principal stress in this area. The azimuth of the maximum horizontal principal stress at different depths is consistent, all along the NWW~SEE direction, between N95 °E and N111 °E. This shows that the location of the new shaft of one Gold Mine is consistent with its geological structure and the orientation of the maximum principal stress measured by many scientific institutions above its -950 m elevation. The maximum horizontal principal stress ratio to the principal vertical stress (lateral pressure coefficient) in the area is between 1.01 and 1.43, which is consistent with the distribution law of the lateral pressure coefficient in China. The test results show that at a depth bigger than 1050 m, the increase of vertical principal stress is equal to that of minimum horizontal principal stress, i.e. about 3 MPa per 100 m. When the depth achieves 1350 m, the vertical principal stress increases by about 3 % per 100 m, and the self-weight stress and the maximum horizontal principal stress grow steadily at about 3 MPa per 100 m. Moreover, the test results make it possible to check the ground pressure monitoring system in each mine. The deep ground pressure appearance was depicted too as well as control measures of deep ground pressure including supporting. The investigation results show that in the roadway and stope situated deeper than 800 m rock mass instability occurred in some mines, and in about 5% of mines low energy rock bursts take place, There was a high-stress area in the lower part of the old goaf, while the area of normal production and construction was stable, without an excessive ground pressure. The risk of rock bursts in deep mines is shortly depicted using a schematic diagram of the rock burst tendency grade of boreholes varying with depth. The evaluation result of rock burst tendency based on the drilling core of the investigated mine is instructive in evaluating rock burst tendency in the region. The typical monzonitic granite, granite, and sericite have more than moderate rock burst tendencies. With the increase of burial depth, the corresponding intensity grade of rock bursts will increase. The authors present problems existing in the mines such as 1) The deep development system is complex, 2) Lack of rock mechanics data in most deep mines, and 3) Some support engineering in deep mining is unreasonable and not in place, 4) The ground pressure monitoring system lacks maintenance, and the early warning ability is poor, 5) The temperature in the deep mine is high, the humidity exceeds the standard, and the ventilation system is not perfect, 6) The intelligence level of the deep mine is low. The authors suggest taking certain activities to reduce, monitor and prevent the safety risk.

The article is divided into 6 sections and 6 subsections: 1. Introduction; 2. Basic Situation of Mines: 2.1. Geological Condition, 2.2. Production Situation; 3. Present Situation of Deep Ground Pressure in Non-coal Mines: 3.1. Regional Tectonic Stress Level, 3.2. Deep Ground Pressure Appearance Sample and parameters selection, 3.3. Control Measures of Deep Ground Pressure, 3.4. Risk of Rockburst in Deep Mines; 4. Problems Existing in the Mines; 5. Prevention and Control Countermeasures and Suggestions; 6. Conclusion. There are 11 figures and 2 tables in the article.

The Authors performed analyses and try to fill part of the gap in the information about the monitoring of ground pressure and rock burst hazard in deep gold mines in China. The most important problems in deep mines were noted and shortly depicted. Some measures to improve the situation were presented.

Response:

Thanks for your summary of the paper.

We noted and depicted the most important problems in deep mines and presented some measures. Thank you for your recognition of these work. We will fill the gaps in the existing work in the follow-up study and conduct more in-depth research. We hope that these works are instructive to the field production.

The article is not scientific enough. There are a lot of depictions, suggestions, and tips which do not make the article scientific. The Authors should keep the introduction comprehensible to scientists working outside the topic of the paper. The conclusion is not good enough. It should be corrected and include deeper insight into the results and their usability. The materials and methods were not depicted. There is no information about tests made on the cores. How were they conducted? How many tests? What was measured? What are the regressions calculated for? What about the rock burst hazard? How the tendency was assessed?

Response:

Thank you so much for this constructive suggestion.

Your comments make us note that this article focuses on the overall evaluation and discussion of gold mines, so there is a lack of description of the experiment. We have added the description of the coring experiment to improve the readability of the paper. Your suggestion makes us realize the importance of increasing the number of boreholes and improving the experimental method. In the future, we will add more parameter analysis and research to enrich the relevant research.

There is a kind of chaos throughout the whole article, which needs to draw the Authors’ attention. The article was written without enough care, which should be corrected.

The design of the article is not clear and logical enough. More consistency would be appreciated and would improve the article’s quality.

Response:

Thanks for your comments.

We carried out an investigation on the safety situation of gold mines, selected the most representative Xiling mining area of Sanshandao in the northwest of Jiaozhou for research, and selected the drilling location according to the principles determined by previous studies. The extracted cores were tested to determine the regional tectonic stress level.

We paid more attention to mine safety production and related facilities in the process of investigation, therefore, the article is more inclined to make a horizontal and comprehensive comparison and description of the safety production situation of each mine, rather than the specific plans and countermeasures of a certain mine. Thank you for pointing out the shortcomings of this paper, and we will conduct a more in-depth and detailed analysis in the follow-up research.

Moreover, English should be improved. There are a lot of grammar errors for example inconsistency between an agent and a verb, the syntax is also not good enough, word order and choice are poor as well as a lot of wordiness. The best way to improve English is by a native speaker consulting.

Response:

Thank you so much for these constructive comments.

We find that some of the sentences in this paper are not rigorous enough or do not conform to the norms, and now the full text has been rewritten and embellished, hoping that these works can improve the quality of the paper.

Round 2

Reviewer 1 Report

NIL

Reviewer 2 Report

It has been modified according to the suggestions, and it is recommended to accept.

Reviewer 4 Report

The article has been improved. Nevertheless, there are no author's own experimental data and the problem is vital only locally. Some English improvement is needed.